Unsupervised inference approach to facial attractiveness

Ibanez-Berganza Miguel miguel.ibanezberganza@gmail.com miguel.berganza@roma1.infn.it 1
Amico Ambra 2
Lancia Gian Luca 1
Maggiore Federico 1
Monechi Bernardo 3
Loreto Vittorio 1 3 4
1 Department of Physics, University of Roma “La Sapienza” , Rome , Italy
2 Chair of Systems Design, Swiss Federal Institute of Technology, Zurich , Switzerland
3 SONY Computer Science Laboratories , Paris , France
4 Complexity Science Hub , Vienna , Austria
Vessel Edward
Electronic publication date: 2020 Oct 28
Publication date: 2020
Volume: 8
Electronic Location ID: e10210
Received 2020 Apr 6; Accepted 2020 Sep 28
Copyright: ©2020 Ibanez-Berganza et al.
Copyright year: 2020
Copyright holder: Ibanez-Berganza et al.
License: This is an open access article distributed under the terms of the Creative Commons Attribution License, which permits unrestricted use, distribution, reproduction and adaptation in any medium and for any purpose provided that it is properly attributed. For attribution, the original author(s), title, publication source (PeerJ) and either DOI or URL of the article must be cited.
License URL: https://creativecommons.org/licenses/by/4.0/

Keywords: Statistical inference, Statistical learning, Facial perception

Funding: The authors received no funding for this work.

==============================
The perception of facial attractiveness is a complex phenomenon which depends on how the observer perceives not only individual facial features, but also their mutual influence and interplay. In the machine learning community, this problem is typically tackled as a problem of regression of the subject-averaged rating assigned to natural faces. However, it has been conjectured that this approach does not capture the complexity of the phenomenon. It has recently been shown that different human subjects can navigate the face-space and “sculpt” their preferred modification of a reference facial portrait. Here we present an unsupervised inference study of the set of sculpted facial vectors in such experiments. We first infer minimal, interpretable and accurate probabilistic models (through Maximum Entropy and artificial neural networks) of the preferred facial variations, that encode the inter-subject variance. The application of such generative models to the supervised classification of the gender of the subject that sculpted the face reveals that it may be predicted with astonishingly high accuracy. We observe that the classification accuracy improves by increasing the order of the non-linear effective interaction. This suggests that the cognitive mechanisms related to facial discrimination in the brain do not involve the positions of single facial landmarks only, but mainly the mutual influence of couples, and even triplets and quadruplets of landmarks. Furthermore, the high prediction accuracy of the subjects’ gender suggests that much relevant information regarding the subjects may influence (and be elicited from) their facial preference criteria, in agreement with the multiple motive theory of attractiveness proposed in previous works.

Introduction

Human facial perception (of identity, emotions, personality dimensions, attractiveness) has been the subject of intense and multidisciplinary research in the last decades (Walker & Vetter, 2016; Little, Jones & DeBruine, 2011; Leopold & Rhodes, 2010). In particular, facial attractiveness is a research topic that involves many different disciplines, from evolutionary biology and psychology to neuroscience (Bzdok et al., 2011; Hahn & Perrett, 2014; Laurentini & Bottino, 2014; Little, 2014; Thornhill & Gangestad, 1999). Furthermore, it is an interesting case of study in the machine learning research community, as a paradigm of a complex cognitive phenomenon, ruled by complex and difficult to infer criteria. Indeed, the rules according to which a facial image will probably result pleasant are poorly known (Little, Jones & DeBruine, 2011). The most relevant face-space variables in terms of which such rules should be inferred remain elusive as well (Laurentini & Bottino, 2014).

Many works have discussed, in the context of evolutionary biology, the validity of the so called natural selection hypothesis (Little, Jones & DeBruine, 2011; Rhodes, 2006), according to which the traits that we recognise as attractive are markers of a good phenotypic condition. Along with natural selection, also sexual selection and the handicap principle are known to play a role in facial attractiveness (Thornhill & Gangestad, 1999).

The evolutionary approach explains several aspects of the phenomenon, such as the impact in facial attractiveness of facial traits that are known to covary with a good phenotypic condition (averageness, symmetry, secondary sexual traits). Despite the success of the evolutionary approach, it is known that there are aspects of facial attractiveness which elude an evolutionary explanation. The natural selection hypothesis implies that the perception of attractiveness is mainly universal, species-typical. While a certain degree of universality has been assessed in many references, cultural and inter-person differences definitely play a role, beyond the species-typical criterion (Little, 2014). Several factors are known to influence the single subject idiosyncrasies, such as the subject’s self- and other-rated attractiveness, genetic propensity, sexual orientation, and the menstrual cycle (see references in Oh, Grant-Villegas & Todorov, 2020).

Recently, many works have argued (the multiple motive hypothesis) that the evaluation of facial attractiveness is a complex process, influenced by the prior inference of semantic personality traits (such as dominance, extroversion or trustworthiness) that we consensually attribute to specific shape and luminance patterns in others’ face (Oh, Dotsch & Todorov, 2019; Oh, Grant-Villegas & Todorov, 2020; Abir et al., 2017; Walker & Vetter, 2016; Adolphs et al., 2016; Galantucci et al., 2014; Little, 2014; Todorov & Oosterhof, 2011; Oosterhof & Todorov, 2008; Edler, 2001; Cunningham et al., 1995). According to this scenario, facial attractiveness is influenced by the single-subject relative inclination towards some fundamental personality traits. In the word of Oh, Dotsch & Todorov (2019), individuals who highly value a personality trait, such as dominance, are likely to perceive faces that appear to possess the trait as attractive. This implies, in particular, that (A) the single subject preferred faces are expected to be, to some extent, distinguishable if characterised or inferred with sufficient accuracy, and (B) they are expected to reflect meaningful information regarding the subject.

The assessment of the validity of these hypotheses is, arguably, strongly influenced by the experimental precision with which the individuals’ preferred faces can be characterised. While the natural selection hypothesis explains general aspects of facial attractiveness, if the experiments allow to resolve the single subjects’ idiosyncrasies, more complex aspects and a strong subjectivity emerge (Hönekopp, 2006; Oh, Grant-Villegas & Todorov, 2020). In particular, the subjectivity of facial attractiveness has been proven to be underestimated by the common experimental method from which most of the works draw their conclusions: the subject-averaged rating assigned to several natural facial images (Hönekopp, 2006). Moreover, it has been argued that the average rating may suffer, as an experimental technique, the curse of dimensionality (the face-space being highly dimensional) and may consequently hinder the complexity and subjectivity of the phenomenon (Laurentini & Bottino, 2014; Valentine, Lewis & Hills, 2016; Ibáñez-Berganza, Amico & Loreto, 2019). Roughly half of the variance in attractiveness ratings has been attributed to idiosyncratic preferences, the other half to shared preferences (Hönekopp, 2006). It is a natural question whether such idiosyncratic proportion would result more prominent using an experimental method that bypasses the use of ratings.

This motivates the search for alternative experimental approaches. Ibáñez-Berganza, Amico & Loreto (2019) investigated the question (A) by means of an innovative experimental technique which permits the sampling of a single subject’s preferred region in the face-space.1 As a matter of fact, the method allows the sampling of the subjects’ preferred facial modifications with high precision. It is observed that, within the experimental precision (limited by the time that the subjects dedicate to the experiment), different subjects “sculpt” distinguishable facial modifications. Indeed, when repeating the experiment, they tend to sculpt facial modifications which are more similar to the ones that they already sculpted than to those sculpted by others, in ∼82% of the cases.

In the present work we present an inference analysis of the data collected in reference (Ibáñez-Berganza, Amico & Loreto, 2019), developing data-driven probabilistic generative models that describe the inter-subject fluctuations around the average landmark positions. As previously stated, such fluctuations are expected to reflect and encode meaningful differences among experimental subjects. Indeed, we thereafter apply such models to the investigation of the aforementioned prediction (B); that is, whether we could elicit meaningful information regarding the subject from her/his preferred facial modifications. In particular, we address whether one may correctly predict the gender of the sculpting subject from such data. Many references have reported quantitative differences between male and female perception of facial attractiveness. Generally, males prefer smaller lower face area, higher cheekbones, larger mouths and eyes (see references in Little, Jones & DeBruine, 2011; Rhodes, 2006; Thornhill & Gangestad, 1999). These facts are compatible with the results of Ibáñez-Berganza, Amico & Loreto (2019). In the present work we go one step further and demonstrate that gender, besides having an impact on the subject-averaged facial sculptures, can actually be predicted with almost certainty for single subjects, based on their facial modifications.

Our inference protocol allows the assessment of the relative influence of linear and non-linear correlations among facial coordinates in the classification, hence overcoming the black-box issue. In particular, we infer, in an unsupervised way, a collection of probabilistic generative models from the database of sculpted facial modifications S=fss=1S (where s is the subject index). Afterwards we assess the predictive power of the models, in two ways. (1) We evaluate the consistency of the simplest of such models, in order to ensure that all of them provide a faithful and economic description of the data. (2) The models are applied, when inferred from male/female data separately, to the supervised classification of the subject gender of a test-set and the results are compared with a powerful, specific algorithm for supervised classification, the random forest algorithm.

We infer, in particular, probabilistic models, Lf|θ, representing the probability density of a facial image with face-space vector f to be sculpted by any subject (given the reference facial portrait and the sculpture protocol). We have considered three generative models of unsupervised learning: two Maximum Entropy (MaxEnt) models, with linear and non-linear interactions among the facial coordinates, and the Gaussian Restricted Boltzmann Machine (GRBM) model of Artificial Neural Network (ANN).2

The models presented here are interpretable: the model parameters θ provide information regarding the relative importance of the various facial distances and their mutual influence in the cognitive process of face perception, which are fundamental questions in the specific literature (Laurentini & Bottino, 2014). In particular, a comparison of the various models’ efficiency highlights the relevance of the nonlinear influence (hence beyond proportions) of facial distances. Finally, this work provides a novel case of study, in the field of cognitive science, for techniques and methods in unsupervised inference and, in particular, a further application of the MaxEnt method (Jaynes, 1957; Berg, 2017; Nguyen, Zecchina & Berg, 2017), otherwise extensively used in physics, systems neuroscience and systems biology (Lezon et al., 2006; Schneidman et al., 2006; Shlens et al., 2006; Bialek & Ranganathan, 2007; Tang et al., 2008; Weigt et al., 2009; Roudi, Aurell & Hertz, 2009; Tkacik et al., 2009; Stephens & Bialek, 2010; Mora et al., 2010; Morcos et al., 2011; Bialek et al., 2012; Martino & Martino, 2018).

The structure of the article is as follows. The inference models will be presented in ‘Materials and Methods’, along with some key methodological details. In ‘Results’ we analyse the results following our objectives (1,2) described above: we first evaluate the quality of our models as generative models of the set of facial modifications in Ibáñez-Berganza, Amico & Loreto (2019). Afterwards, we perform a further assessment in which we apply the generative models to the classification of the gender of the subjects from their sculpted facial vectors, and compare the results with that of a purely supervised learning algorithm. We draw our conclusions in ‘Conclusions’.

Materials and Methods

Description of the database

We analyse the dataset S described by Ibáñez-Berganza, Amico & Loreto (2019). In such experiments, each subject was allowed to sculpt her/his favorite deformation of a reference portrait (through the interaction with an software which combines image deformation techniques with a genetic algorithm for the efficient search in the face-space). The set of selected images are, hence, artificial, though completely realistic, variations of a common reference portrait (corresponding to a real person). In such a way, only the geometric positions of the landmarks are allowed to vary, the texture degrees of freedom are fixed (and correspond to the reference portrait RP1 (taken from the Chicago database, see (Ma, Correll & Wittenbrink (2015)) and Fig. 1). This representation of the face is, hence, rooted on a decoupling of geometric (also called shape) and texture (also called reflectance) degrees of freedom (Laurentini & Bottino, 2014).3 See Description of the E1 experiment in the Supplemental Information for a more detailed descriptions of the experimental protocol in Ibáñez-Berganza, Amico & Loreto (2019).

Figure 1 Facial landmarks i = 0, …, 7 whose 2D coordinates r→i constitute the face space (signaled with black circles). Their position in the figure correspond to the average position, 〈r→i〉. The background image corresponds to the texture degrees of freedom of the reference portrait. The blue lines are polar histograms h(ϕ) (the radius is proportional to h(ϕ)) corresponding to the experimental distribution of angle landmark fluctuations around their average position.

The database consists in the set of landmark geometric coordinates S=rss=1S, where s is the facial vector index corresponding to the N=28 vectors sculpted by each of the ns = 95 experimental subjects (hence: S=nsN=2,660).4 We will call rs=rx,1s,…,rx,ns,ry,1s,…,ry,ns the vector whose 2n components are the (x, y) Cartesian coordinates of a set of n = 8 landmarks, in units of the facial height. Such landmarks (those signaled with an empty circle in Fig. 1) are a subset of the set of landmarks used for the image deformation in Ibáñez-Berganza, Amico & Loreto (2019) (signaled with red points in Fig. 1). We will also refer to the 2D Cartesian vector of the ith landmark as r→i=xi,yi, and define the fluctuations of the landmark positions with respect to their average value as Δ→i=r→i−〈r→i〉, where 〈⋅〉 denotes the experimental average, 〈⋅〉 = (1∕S)∑s⋅. Analogously, Δ(s) = r(s) − 〈r〉. An important aspect of the dataset is that even the coordinates of the restricted set of n = 8 landmarks, r(s), are redundant and depend on 10 coordinates only, due to the presence of 2n − 10 = 6 constraints that result from the very definition of the face-space. Such constraints are described in detail in the Supplemental Information.

Unsupervised inference

We now present some non-technical notion of unsupervised learning. In the next subsection we will describe the probabilistic models with which we describe the dataset. These are generative models that induce a likelihood probability density L⋅|θ over the space of facial vectors Δ. The meaning of LΔ|θ is that the probability of finding a facial vector in an interval of facial vectors I is given by ∫ILΔ|θdΔ. The probabilistic model L represents a generalisation of the database, from which it is not unambiguously elicited. Indeed, a probabilistic model of the data depends both on the functional form of L (often called simply the model), and on the learning algorithm, or the protocol with which its parameters θ are inferred form the data.

For the learning algorithm, in the present work we adopt the Maximum Likelihood principle. We fix the parameters to the value that maximises the database likelihood: θ∗= argmaxθ∏sLΔs|θ, where the product is over all the samples in the database (please, see Inter- and intra-subject correlations and errors in the Supplemental Information for further information at this regard).

The functional form is given by the kind of unsupervised model. In this article we will consider three models (2- and 3-MaxEnt, GRBM), that will be described in the following subsections. The 2- and 3-MaxEnt models follows from the Maximum Entropy principle (Jaynes, 1957; Berg, 2017; Nguyen, Zecchina & Berg, 2017; Martino & Martino, 2018), which provides the functional form of the probability distribution L⋅|θ that exhibits maximum entropy and, at the same time, is consistent with the average experimental value of some observables of the data, 〈Σ〉, that will be called sufficient statistics. L must satisfy 〈Σ〉L=〈Σ〉, where 〈⋅〉L refers to the expected value according to L. In other words, L is the most general probability distributions constrained to exhibit a fixed expectation value of 〈Σ〉L (and this value, under the Maximum Likelihood prescription, is given by the corresponding experimental value). The precise choice of the sufficient statistics determines the functional form of LΔ|θ. A more detailed description is given in the next section and in section Introduction to the Maximum Entropy principle: correlations vs effective interactions of the Supplemental Information.

The maximum entropy models

We propose two MaxEnt probabilistic generative models of the set of selected faces, inferred from the dataset S. In the case of the Gaussian or 2-MaxEnt model, the sufficient statistics is given by the 2n averages 〈Δμ〉 and by the 2n × 2n matrix of horizontal, vertical and oblique correlations among couples of vertical and horizontal landmark coordinates, whose components are Cμν = 〈ΔμΔν〉. In these equations, the 2n Greek indices μ = i, ci denote the ci = x, y coordinates of the ith landmark. The 2-MaxEnt model probability distribution takes the form (see Supplemental Information) of a Maxwell–Boltzmann distribution, LΔ|θ=1Z exp−HΔ|θ. In this equation, Z is a normalising constant (the partition function, in the language of statistical physics) depending on θ, and H = H2 (the Hamiltonian) is the function: (1) H2Δ|θ2=12Δ†⋅J⋅Δ+h†⋅Δ.

The model depends on the parameters θ2 = {J, h}, or the 2n × 2n matrix of effective interactions J and the 2n vector of effective fields, h. Due to the symmetry of matrix J, the number of independent parameters in the 2-MaxEnt model is D + D(D + 1)∕2, where D = 2n is the dimension of the vectors of landmark coordinates Δ. The value of these parameters is such that the equations 〈Δ〉=〈Δ〉L and 〈ΔμΔν〉L=Cμν are satisfied. This is equivalent to require that θ2 are those that maximise the likelihood of the joint L over the database S (the Maximum Likelihood condition). The solution of such an inverse problem is (see Supplemental Information): J = C−1, h = J⋅〈Δ〉, and Z = (2π)nexp(h†⋅J−1⋅h∕2)(detJ)−1∕2, where the −1 power in equation J = C−1 denotes the pseudo-inverse operation, or the inverse matrix disregarding the null eigenvalues induced by the database constraints (see Supplemental Information). The 2-MaxEnt model is equivalent in the present case, in which the coordinates Δμ are real numbers, to a Principal Component Analysis and L is in this case a multi-variate Normal distribution.

We will define as well the 3-MaxEnt model. In this case, the sufficient statistics is given by averages 〈Δμ〉, pairwise correlations Cμν, and correlations among 3-landmark coordinates, Cμνκ3=〈ΔμΔνΔκ〉. The 3-MaxEnt model probability distribution function takes the form of a Maxwell–Boltzmann distribution multiplied by a regularisation term ensuring that it is normalisable: (2) L⋅|h,J,Q=1Z3e−H2⋅|θ2+H3⋅|QH⋅|B

where H(⋅|B) is a multivariate Heaviside function, equal to one for vectors Δ lying in the hypercube −B ≤ Δμ ≤ B for all µand zero otherwise; Z3 is the normalising factor, and the Hamiltonian is H = H2 + H3, where H2 given by Eq. (1) and H3 by: (3) H3Δ|Q=16∑μνκΔμΔνΔκQμνκ

Besides h and J, the non-linear MaxEnt model depends on a further tensor of three-wise interaction constants among triplets of landmark coordinates. Consequently, the number of independent parameters is [D] + [D(D + 1)∕2] + [(D3 − D2)∕6 + D] = D3∕6 + D2∕3 + 5D∕2.

The solution of the inverse problem for the non-linear MaxEnt model does not take a closed analytic form. The maximum likelihood value of the parameters (h, J, Q) is numerically estimated by means of a deterministic gradient ascent algorithm (see Monechi, Ibáñez-Berganza & Loreto, 2020). A detailed explanation of the learning protocol may be found in the Supplemental Information (see Learning in the non-linear MaxEnt model). Before inferring the data with the non-linear models (3-MaxEnt and GRBM) we have eliminated a subset of 6 redundant coordinates from the original 2n coordinates. The data has been standardised in order to favor the likelihood maximisation process. The value of B has been chosen to be B = 6, so that the probability distribution function is nonzero only in an hypercube whose side is six times the standard deviation of each standardised variable.

The Restricted Boltzmann Machine model for unsupervised inference

We have learned the data with the (Gaussian-Binary) Restricted Boltzmann Machine (GRBM) model of unsupervised inference (Wang, Melchior & Wiskott, 2012; Wang, Melchior & Wiskott, 2014). It is a 2-layer unsupervised ANN, a variant, processing input real vectors, of the binary-binary RBM model. The model induces a probability distribution LΔ|θ=∑hpΔ,h|θ which is obtained by the marginalisation, over a set of Nh binary hidden variables (or hidden neurons), hj ∈ {0, 1}, j = 1, …, Nh, of a joint probability distribution p: (4) LΔ|θ= ∑hpΔ,h|θ,pv,h|θ=1Zθe−Ev,h|θ

The interaction among visible and hidden variables, and the dependence of p(Δ, h|θ) on all its arguments is described by an energy function E that couples hidden to visible neurons. E is defined in terms of a set of parameters θ consisting, among others, on the D × Nh matrix of synaptic weights among visible (input) and hidden variables. Although E presents only a linear coupling among v and h, the marginalisation over binary hidden neurons actually induce nonlinear effective couplings at all orders among the visible variables v (or Δ), couplings that may be accessed from the network parameters θ (MacKay, 2003; Cossu et al., 2019).

We have employed the open-source software (Melchior, 2017) for the efficient learning of GRBM. The learning protocol and parameters are described in detail, along with an introduction to the GRBM model, in the Supplemental Information, see: Learning the database with the Gaussian Restricted Boltzmann Machine.

Results

We will now present an assessment of the description of the database according to the inference models described in the precedent section. In ‘Quality of the MaxEnt models as generative models’ we will argue that the 2-MaxEnt model is a faithful representation of the dataset, and that only the nonlinear models predict the subject’s gender when applied to such supervised inference task. Finally, in ‘Analysis of the matrix of effective interactions’, we will argue that the matrix of effective interactions J provides interpretable information, beyond the raw information present in the raw experimental measure C.

Quality of the MaxEnt models as generative models

Histograms of single landmark-angle fluctuations

The quality of the 2-MaxEnt generative model as a faithful description of the database may be evaluated by the extent to what the model L reproduces observables O that it is not required to reproduce by construction. In other words, observables that cannot be written in terms of couples and triplets of coordinates Δα. The model is faithful in the extent to what 〈O〉≃〈O〉L.

The ith landmark coordinates Δ→i tend to fluctuate in the database with respect their average position 〈Δ→i〉=0→. As a nonlinear observable O we will consider the angle that the ith landmark fluctuation Δ→i forms with the x-axis. This quantity will be referred to as ϕis= arctanΔi,ys∕Δi,xs. In Figs. 1 and 2, we report the empirical histogram of angles, h(ϕi) for some landmarks i. Remarkably, some landmarks’ angle distribution exhibit local maxima, probably reflecting their tendency to follow the direction of some inter-landmark segments (as it is apparent for the 3-rd and 6-th landmark’s in Fig. 1).5 We have compared the empirical histograms with the theoretical ϕi distributions according to the model. These have been obtained as the angle histograms of a set of S vectors Δ sampled from the inferred distribution L⋅|θ (see Fig. 2). The 2-MaxEnt model satisfactorily reproduces most of the landmark angle distributions. The empirical angle distribution, in other words, is reasonably well reproduced by the theoretical distribution hφ= ∫dΔLΔ|θδϕΔ−φ.

Figure 2 Comparison among empirical (h(i)(ϕ)) and theoretical (htiϕ) histograms of angle landmark fluctuations, for several landmarks, i = 1, 5, 6, 7, see Fig. 1 (from left to right, from top to bottom). The empirical histograms h(i)(ϕ) represent the probability density of empirical displacements Δ→is of the ith landmark along an axis which subtends an angle ϕ ∈ (−π, π) with the horizontal axis. These histograms are presented for all the landmarks also in Fig. 1, under the form of polar histograms. htiϕ is the theoretical prediction of the same quantity, obtained by sampling data from the inferred L⋅|θ.

It is important to remark that the model-data agreement on h(ϕi) is observed also for large values of |ϕi| ∈ (0, π) (see Fig. 2), and not only for small values of |ϕi|, for which it approximately becomes |Δi,y∕Δi,x| (whose average is related to the correlation Cαβ, see Supplemental Information).

We conclude that, very remarkably, a highly non-linear observable as ϕ is well described by the 2-MaxEnt model, albeit it has been inferred from linear (pairwise) correlations only. In this sense, the 2-MaxEnt model is a faithful and economic description of the dataset. This picture is confirmed by the results of the following section which suggest, however, that a description of the gender differences in the dataset require taking into account effective interactions of order p > 2.

Performance of the MaxEnt model in a classification task

We now further evaluate the quality of the 2- and 3-MaxEnt models by assessing their efficiency to classify a test database of vectors in two disjoint subsets S=SA∪SB corresponding to the gender of the subject that sculpted the facial vector in Ibáñez-Berganza, Amico & Loreto (2019). We compare such efficiency with that of the GRBM model of ANN (see Wang, Melchior & Wiskott (2012) and Wang, Melchior & Wiskott (2014), ‘Materials and Methods’ and the Supplemental Information). This comparison allows to assess the relative relevance of products of p-facial coordinates Δα in the classification task: averages (p = 1), pairwise correlations (p = 2), and non-linear correlations of higher, p > 2 order (modelled by the 3-MaxEnt and GRBM models only).

The dataset is divided in two disjoint classes SA,SB. Afterwards, both SA,B are divided in training- and test- sets (20% and 80% of the elements of SA,B, respectively), and inferred the A and B training sets separately, with the MaxEnt and GRBM models. This results in six ({2, 3, G} × {A, B}) sets of parameters θA,B2,3,G, where the super-index refers to the model. Given a vector Δ belonging to the A or B test set, the score sΔ= lnLΔ|θA−lnLΔ|θB is taken as the estimation of the model prediction for Δ∈SA. The resulting Receiver Operating Characteristic (ROC) curves (Murphy, 2012) are shown in Fig. 3 for the various models considered.6

Considering only the averages 〈Δ〉 as sufficient statistics (or, equivalently, inferring only the fields h and setting Jij=σi−2δij in Eq. (1)) results in a poor, near-casual classification (specially in the most interesting region of the ROC curve, for small FPR and large TPR), see Fig. 3. The 2-MaxEnt model allows, indeed, for a more efficient classification. Rather remarkably, with the 3-MaxEnt and GRBM models the classification accuracy gradually increases. We interpret this as an indication of the fact that non-linear effective interactions at least of fourth order are necessary for a complete description of the database. For completeness, we have included a comparison with the Random Forest (RF) algorithm (Murphy, 2012). As shown in Fig. 3, RF achieves the highest classification accuracy (auRO C = 0.995, see Supplemental Information). We notice that this does not imply that the unsupervised models are less accurate: the RF algorithm is advantaged, being a specific model trained to classify at best the A, B partitions, not to provide a generative model of the A and B partitions separately.

Figure 3 True Positive Rate (TPR) versus False Positive Rate (FPR) corresponding to the Receiver Operating Characteristic (ROC) curves associated to the gender classification. Different ROC curves correspond to different algorithms. PC’s refers to a t-Student test of the difference in the principal components of a vector with respect to their average value in the A, B sets.

We report the maximal accuracy scores for all the algorithms: RF (0.971); GRBM (0.952); 3-MaxEnt (0.865); 2-MaxEnt (0.764); 1-MaxEnt (0.680). The 2-MaxEnt model efficiency is, as expected, compatible with that of a t-Student test regarding the differences in the principal component values of A and B vectors, see Fig. 3. See the auROC scores of all the algorithms in the Supplemental Information.

We conclude that, on the one hand, the subjects’ gender strikingly determines her/his preferred set of faces, to such an extent that it may be predicted from the sculpted facial modification with an impressively high accuracy score (Murphy, 2012): a 97.1% of correct classifications. On the other hand, the relative efficiency of various models highlights the necessity of non-linear interactions for a description of the differences among male and female facial preference criterion in this database. Arguably, such nonlinear functions play also a role in the cognitive process of facial perception. The criterion with which the subjects evaluate and discriminate facial images seems to involve not only proportions rα∕rβ (related to the pairwise correlations Cαβ, see Supplemental Information), but also triplets and quadruplets of facial coordinates influencing each other (yet, see the Supplemental Information for an alternative explanation).7

It is believed that the integration of different kinds of facial variables (geometric, feature-based versus texture, holistic, see Trigueros, Meng & Hartnett, 2018; Valentine, Lewis & Hills, 2016) improve the attractiveness inference results, suggesting that both kinds mutually influence each other in attractiveness (Eisenthal, Dror & Ruppin, 2006; Xu et al., 2017; Laurentini & Bottino, 2014; Ibáñez-Berganza, Amico & Loreto, 2019). The present results indicate that, even restricting to geometric coordinates (at fixed texture degrees of freedom), it is necessary a holistic approach, in the sense that it considers the mutual influence of many geometric coordinates.

Actually, our results indicate that pairwise influence of geometric coordinates are enough for a fair and economic description of the database (the 2-MaxEnt model predicts nonlinear observables, beyond the empirical information with which it has been fed). However, the differences among the facial vectors sculpted by males and females are not only encoded in pairwise correlations among geometric coordinates. Facial vectors reveal the gender of the sculpting subject with almost certainty only when non-linear models are used.

Analysis of the matrix of effective interactions

We now show that the generative models may provide directly interpretable information. This is an advantage of the MaxEnt method, whose parameters, the effective interaction constants, may exhibit an interpretable significance. We prove, in particular, that at least the matrix of effective interactions J admits an interpretation in terms of “resistance” (elastic constant) of inter-landmark segment distances and angles to differ with respect to their average or preferred value.

The 2-MaxEnt model admits an immediate interpretation. The associated probability density LΔ|θ= exp−H2Δ|θ∕Z formally coincides with a Maxwell–Boltzmann probability distribution of a set of n interacting particles in the plane (with positions Δ→i, i = 1, …, n), subject to the influence of a thermal bath at constant temperature. Each couple i, j of such fictitious set of particles interacts through an harmonic coupling that corresponds to a set of three effective, virtual springs with non-isotropic elastic constants, J(xx)ij, J(yy)ij, J(xy)ij corresponding (see Eq. (1)) to horizontal, vertical and oblique displacements, Δi,x − Δj,x, Δi,y − Δj,y, and Δi,x − Δj,y, respectively.

The inferred effective interactions are more easily interpretable if one considers, rather than their x x, y y and x y components, the longitudinal and torsion effective interactions, Jij∥ and Jij⊥, respectively. The longitudinal coupling |Jij∥| may be understood (see the Supplemental Information for a precise definition) as the elastic constant corresponding to the virtual spring that anchors the inter- ij landmark distance to its average value, 〈rij〉 (where r→ij=r→j−r→i). In its turn, the torsion interaction |Jij⊥| is the elastic constant related to fluctuations of r→ij along the direction normal to r→ij or, equivalently, to fluctuations of the ij-segment angle, with respect to its average value that we will call αij = arctan(〈rij,y〉∕〈rij,x〉).

In Figs. 4A, 4B we show the quantities |Jij∥| and |Jij⊥| for those couples i, j presenting a statistically significant value (for which the t-value tij = |Jij|∕σJij > 1 (a description of the calculation protocol of the bootstrap error σij may be found in the Supplemental Information). The width of the colored arrow over the i, j segment is proportional to |Jij∥| (blue arrows in Fig. 4A) and |Jij⊥| (red arrows in Fig. 4B). We notice that there exist inter-landmark segments for which |Jij∥| is significant while |Jij⊥| is not (as the 0, 4 or the 5, 6 segments) and vice-versa (as the 6, 7 and 2, 5). This suggests that |Jij∥|, |Jij⊥| actually capture the cognitive relative relevance of distance fluctuations around 〈rij〉, and of angle fluctuations around αij.

Figure 4 Modulus of the matrices J∥ (A), J⊥ (B), C∥ (C). The width of the arrow joining the ith and jth landmarks is proportional to |Aij|, where A is the corresponding matrix. Matrices J∥ and J⊥ represent, respectively, the longitudinal and torsion elastic constants of the correspondent inter-landmark segments. They indicate, respectively, the segment’s distance and angle “resistance” to differ in the database with respect to their average (preferred) values, reported in the image. Only significant matrix elements have been plotted: only those exhibiting a t-value larger than one: tij = |Aij|∕σAij > 1. Matrix C∥ (C) is less interpretable than J∥ (A).

We remark that the prominent importance of the inter-segment angles ij highlighted in Fig. 4B is fully compatible with the analysis presented in (Ibáñez-Berganza, Amico & Loreto, 2019) at the level of the oblique correlation matrix C(xy), and it goes beyond, as far as it quantitatively assess their relative relevance. As we will see in the next subsection, such information cannot be retrieved from the experimental matrix C only.

In the Supplemental Information we explain in more detail the analogy with the system of particles. We also analyse the dependence of the torsion and longitudinal effective interactions, |Jij∥| and |Jij⊥|, with the average distance and angle of the ij inter-landmark segment, showing that there is a moderate decreasing trend of |Jij∥| with 〈rij〉. This fact admits an interpretation: large inter-landmark distances are less “locked” to their preferred value with respect to shorter distances. Very interestingly, such trend is less evident for |Jij⊥|: the relative relevance of the inter-landmark angles according to J is not lower for farther away landmarks. This confirms the holistic nature of facial perception. The mutual influence among landmarks is not among nearby landmarks only, but over the scale of the entire face.

Extra information retrieved with effective interactions

A relevant question is to what extent the inferred effective interactions J provide interpretable information, inaccessible from the raw experimental correlations C. In the general case, couples of variables may be statistically correlated through spurious correlations, even in the absence of a causal relation among them (see the Introduction to the Maximum Entropy principle in the Supplemental Information). In the present case, the main source of spurious correlations is the presence of the constraints among various landmark coordinates. The MaxEnt inference eventually subtracts (through the pseudo-inverse operation) the influence of the constraints from matrix J, which describes the essential effective mutual influence among pairs of coordinates of prominent relative importance (see Two ways of inferring with constraints in the database of facial modifications in the Supplemental Information).

The differences among C and J matrices are apparent from Fig. 4C, where the arrows represent the absolute value of the raw experimental matrix elements Cij∥. Remarkably, all but two of the matrix elements result statistically significant (t-value >1): matrix C∥ can be hardly used to assess the relative relevance of various inter-landmark segments. The effective interaction matrix J disambiguates the correlations propagated by the constraints, attributing them to the effect of a reduced set of elastic constants, in the particle analogy. Such attribution is not unambiguous, but the result of the inference procedure.

In other words, the inferred matrix J provides interpretable results, beyond the less-interpretable empirical information present in matrix C. Indeed, matrix C is dense, in the sense that almost all its elements are statistically significant (see Fig. 4C). Matrix C is, however, exactly explained by the 2-MaxEnt model, defined by the sparser matrix J of effective interactions, such that only some matrix elements are statistically significant (see Figs. 4A, 4B).

An in-depth comparison among C and J is presented in the Supplemental Information, where we consider also the alternative method of avoiding the constraints, consisting in inferring from a non-redundant set of coordinates. We conclude that, in the general case, and for the sake of the interpretation of the effective interactions, it may be convenient to infer from a database of redundant variables, eliminating a posteriori the influence of the null modes associated to the constraints (and, perhaps, of low-variant modes associated to quasi-constraints or to non-linear constraints).

Conclusions

We have performed an unsupervised inference study of the database of preferred facial modifications presented in reference (Ibáñez-Berganza, Amico & Loreto, 2019). Much work has been devoted to the regression of the average rating in face-space, specially in the machine learning community. Such supervised inference approach indirectly allows for an assessment of the relative impact of various facial traits on perceived attractiveness. This point, however, remains poorly understood (Laurentini & Bottino, 2014). Furthermore, some authors have argued that the subjective nature of facial attractiveness has been overlooked and underestimated, and that the subject-averaged rating presents several limitations as an experimental method (Hönekopp, 2006; Laurentini & Bottino, 2014; Oh, Grant-Villegas & Todorov, 2020). As a novel experimental tool for the investigation of the nature of facial attractiveness and its subjectivity, we here propose an alternative inference scheme in which the variability to be inferred is the inter-subject variability of preferred modifications in a subspace of the face-space (in which only the geometric positions of some landmarks are allowed to vary), rather than the average rating assigned to different natural faces.

The present work is probably the first unsupervised inference approach in facial preference research. Our models induce a probabilistic representations of a set of facial modifications (corresponding to the whole set of subjects or to the set of male or female subjects only, in ‘Histograms of single landmark-angle fluctuations’, ‘Performance of the MaxEnt model in a classification task’ respectively, or to the single subject). Such models avoid the use of ratings; account for nonlinear influence of p −plets of facial features, hence beyond a principal component analysis; their parameters are in principle interpretable since they involve “physical” facial coordinates only.

Our approach allows to clarify several aspects regarding facial preference. First, that the cognitive mechanisms related to facial discrimination in the brain mainly involves proportions, or pairwise influence of couples of landmarks, more than the positions of single landmarks. Indeed, the 2-MaxEnt model, equivalent to a description in terms of principal components, is enough to describe also non-linear features of the database (see ‘Histograms of single landmark-angle fluctuations’). Moreover, the results suggest as well that non-linear operators of the geometrical facial coordinates may play a non-negligible role in the cognitive process (see ‘Performance of the MaxEnt model in a classification task’). Further research is needed to clarify this point (see also Cognitive origin of non-linear correlations in the Supplemental Information).8

Second, and rather remarkably, the introduction of non-linear effective interactions, beyond the influence of proportions, allows for an astonishingly high classification efficiency of the facial vectors according the subject’s gender. Indeed, the random forest algorithm, a highly nonlinear supervised algorithm for classification, provides a 97% of correct classifications. The most non-linear of the probabilistic models, the GRBM, provides a slightly lower accuracy: 95%. This implies that the subject’s gender strikingly determines her/his facial preference criteria and that the sculpted facial modifications, as a sample of the subjects idiosyncratic criterion, are accurate enough to allow to predict such impact.

The impact of the gender is consistent with the sexual selection hypothesis (Little, Jones & DeBruine, 2011; Rhodes, 2006; Thornhill & Gangestad, 1999). However, since the sculpted vectors partially capture the subjects’ idiosyncrasy, such a result is also consistent with the multiple motive hypothesis, assuming that the gender strongly influences the subjects’ idiosyncratic preferences for personality traits (in the language of Oh, Grant-Villegas & Todorov, 2020).

In summary, we have presented probabilistic generative models of the database of preferred facial variations, describing the inter-subject fluctuations around the average modification (given a reference background portrait). The simplest of these models, characterised by pairwise correlations among facial distances, already provides a faithful description of the database. Afterwords, we demonstrate that such fluctuations encode, and may accurately reveal when introducing non-linearity, the subjects’ gender. According to the multiple motive hypothesis, many other subject attributes and distinguishing psychological traits may influence, beyond the gender, the preferences in the face-space. The present results suggest that such attributes could be retrieved from the subject sculpted facial vectors.

Finally, we have demonstrated that the data elicited with the method in Ibáñez-Berganza, Amico & Loreto (2019) represents a novel case of study for the application of statistical learning methods, in particular the assessment of the relevant order of interaction by comparison with an ANN model, and the comparison among various strategies of inference in the presence of constraints.

The introduction of texture degrees of freedom in the sculpting process is a possible development of the empirical technique of Ibáñez-Berganza, Amico & Loreto (2019), that would allow to quantify the extent to which texture and geometric facial features (and which ones) influence each other in attractiveness perception (a debated question, see Laurentini & Bottino, 2014). Further possible extensions are: the generalisation to different datasets and facial codification methods allowing, for example, landmark asymmetry (see also Generality of the unsupervised inference models in the Supplemental Information); the classification of different subject’s features from her/his set of sculpted faces.

Supplemental Information

Supplemental Information 1 The data analysis software used to produce the results

Jupyter Python scripts and modules used to implement the inference models and to analyse the data.

Click here for additional data file.

Supplemental Information 2 Detailed information on the database, the inference protocol, the error estimation and the interpretation of the results

Includes a theoretical introduction to the Maximum Entropy method and a discussion regarding the MaxEnt inference in the presence of constraints among the vector coordinates.

Click here for additional data file.

Supplemental Information 3 Dataset corresponding to the experiment ”E1” first analysed in Ibanez-Berganza et al, 2019

A detailed description of the database can be found in the README file.

Click here for additional data file.

We acknowledge Andrea Gabrielli, Irene Giardina, Carlo Lucibello, Giorgio Parisi and Massimiliano Viale for inspiring discussions. We give particular thanks to Andrea Cavagna for his suggestions and comments to the draft.

Additional Information and Declarations

Competing Interests

Author Contributions

Data Availability

1 The alternative experimental technique allows a given subject to seek her/his preferred variation of a reference facial portrait. Such variations differ only in a low-dimensional face-space of essential facial features. It is arguably the introduction of these two ingredients: the reduction of facial degrees of freedom and the possibility to efficiently explore the face-space (rather than rating facial images differing in many facial dimensions) that allows for a significant experimental distinction of different subject’s criteria (see the Supplemental Information for further details).

2 This inference scheme differs from the common one found in the facial attractiveness literature, specially in the machine learning papers. In these papers the main goal is the automatic rating of facial images, considered as a supervised inference problem (Laurentini & Bottino, 2014). The facial image is parametrised in a face-space vector f, the inference goal consists in the regression R(f) that reproduces at best the subject-averaged ratings 〈Rs〉s of a database {fs, Rs}. In the case of deep, hierarchical networks, which automatically perform feature selection, the raw image is used as an input to the learning algorithm instead of a face-space parametrisation f. The resulting relevant features are, however, not immediately accessible.

3 Many works exploit the geometric/texture decoupling in artificial facial images to study separately the effect of both kinds of coordinates. It is also a natural strategy of dimensionality reduction of the human face, that has been observed to be implemented in both the neural code for facial identification in the brain and by artificial neural networks (Chang & Tsao, 2017; Higgins et al., 2020). In Ibáñez-Berganza, Amico & Loreto (2019), we combine this separation with the use of completely realistic images, thus eliminating the bias that artificial images are known to induce in experiments (Balas & Pacella, 2015; Oh, Grant-Villegas & Todorov, 2020).

4 Actually, the database S=rv,i is composed by S=ns×N facial vectors labelled by a single index s = 1, …, S or, alternatively, by a tuple of indices (v, i) (v = 1, …, ns, i=1,…,N, ns = 95, N=28) referring to the ith facial vector sculpted by the v-th subject (in a single genetic experiment, see Ibáñez-Berganza, Amico & Loreto, 2019). In the Supplemental Information we present a detailed analysis of the error estimation over the dataset, distinguishing inter- and intra-subject fluctuations. The last ones are, in principle, an artifact of the sculpting process, but they may encode part of the subject’s idiosyncrasy. Similarly, the inferred models may be conceived to account for intra- and inter-subject, or only for inter-subject correlations (see the Supplemental Information).

5 Interestingly, such local maxima seem to be oriented along inter-landmark segments eventually involving landmarks which are not described in the facial vectors Δ→: the landmarks ℓ0 and ℓ18, see the Supplemental Information.

6 These consist in a scatter plot with the fraction of true positive classifications (TPR) in the SA test-set versus the fraction of false positive classifications (FPR) in the SB test-set, where each point corresponds to a different soil δ over the estimator s(Δ)≶δ that we use to assign whether the model predicts that Δ belongs to A or B. The curve is invariant under reparametrizations of s → f(s) defined by any monotone function f.

7 As we explain in the Supplemental Information, the non-Gaussian correlations of order 3 present in the dataset are, at least partially, not of cognitive origin, but due to an artifact of the numerical algorithm allowing subjects to sculpt their preferred facial vectors. However, we believe that the non-linear effective interactions that we infer do reflect the existence of non-linear operations playing a role in the cognitive process of facial evaluation. This is suggested by the fact that the introduction of non-linear effective interactions drastically improves the gender classification.

8 The recent de-codification of the neural code for facial recognition in the primate brain (Chang & Tsao, 2017) has revealed that recognition is based on linear operations (or projections in the geometric and texture principal axes) in the face-space.

Bernardo Monechi and Vittorio Loreto are employed by Sony Computer Science Laboratories.

Miguel Ibanez-Berganza conceived and designed the experiments, performed the experiments, analyzed the data, prepared figures and/or tables, authored or reviewed drafts of the paper, and approved the final draft.

Ambra Amico performed the experiments, analyzed the data, prepared figures and/or tables, and approved the final draft.

Gian Luca Lancia and Federico Maggiore analyzed the data, prepared figures and/or tables, and approved the final draft.

Bernardo Monechi analyzed the data, prepared figures and/or tables, authored or reviewed drafts of the paper, and approved the final draft.

Vittorio Loreto performed the experiments, authored or reviewed drafts of the paper, and approved the final draft.

The following information was supplied regarding data availability:

Data are available in the Supplemental Files.

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
