# Peer review of "Unsupervised inference approach to facial attractiveness"

_PeerJ, doi:10.7717/peerj.10210_

## Round 0.1 · original submission · Major Revisions

The reviewers found the research question interesting, and the approach and results to be promising.

However, the reviewers highlighted a number of issues that need to be addressed in a revision.

All three reviewers, and myself, identified a lack of clarity in the writing that made it difficult at times to make sense of the findings. This was true in defining the motivation for the study (Rev 2) as well as in the methods and results (all 3 reviewers).

Concerns were also raised about the stimuli and task (Rev 1), and about the subjects (Rev 1 and Rev 3).

Myself and the reviewers also identified many grammatical and spelling errors. I would ask the authors to have the manuscript more carefully copyedited.

I would also encourage the authors to revise the manuscript with an eye toward making it more understandable to a less technical audience that is interested in facial attractiveness but may not work directly with these types of generative models. It would also be helpful to make sure that this paper can be understood with minimal reference to Ibáñez-Berganza et al. (2019).

My additional comments:

Figure 3: please include full labels for TPR or FPR in the figure legend.

ln 225-227: I think this calls for reference to the large literature on “holistic” facial representations.

Fig. 4: 3rd panel is not very interpretable

ln 325-327: It is not clear to me that the manuscript benefits from mentioning the possible security or forensic applications of the results. I would consider cutting.

Reviewer 1 ·

Basic reporting

no comment

Experimental design

no comment

Validity of the findings

no comment

Additional comments

Overall, this is an interesting approach. I appreciate the earlier authors’ development of a new method to extract human preference behavior in relation to faces. It’s a clever and fairly naturalistic approach. In the current work, the predictability of gender difference is an interesting and significant addition. However, I see issues with the interpretation of the present results, and also underlying issues and lacunae in the psychophysical procedure.

1. The model introduces symmetry to otherwise asymmetrical faces by applying the same transform to both sides of the face. This is somewhat unavoidable but it has yet to be shown that, given freedom to manipulate both sides of the face, people don’t introduce small asymmetries, if only to make a face less “uncanny.” In any case, the present method does not help disentangle the influence of symmetry. Nor is it clear that people really “like” the sculpted faces they make. I would have had users compare each user-sculpted face to a symmetrized version of the same (unaltered) face—individuals should prefer the face they sculpted (maybe with some control for familiarity). If this is not the case, the MaxEnt models for individuals aren’t very meaningful.
2. Also it is not clear that people can see the changes they make in the sculpted faces.
a. Stimuli are B/W (compressed?) 300x400 pixels, shown at unreported viewing distance, and there seems to be no control for individuals’ optical errors (corrected to normal?). Maybe people selected preferred images in each pair based on visibility of difference compared to the previous trial. In any case, no one would confuse these small, low-res images for real faces.
b. Subjects were apparently uncompensated and mostly undergraduates so it’s not clear how careful or motivated they were. Also, what were the instructions? Choose the face you like best? Or the “choose the most attractive face”? These might be different: the first question is about personal preference while the second could be about cultural norms. (also how is gender assessed? Again there are culturally normative effects possible here).
3. With the generative models, why not generate new sculpted faces tailored for each subject and have that subject choose between that face and, e.g., a face sculpted from the preferences of someone else of the same gender? The subject should choose the face sculpted from their own preferences. As it is, the generative model could just be overfitting an individual’s idiosyncracies of adjustment rather than preference (except for the gender difference).
4. Showing model accuracy for gender of the observer doesn’t really follow or support the premise of the work, which is that individual preferences are systematic. I would refocus on the gender result rather than claiming that you are modeling individual preference.

Minor comments

Lots of grammar issues. A non-exhaustive list:

Citations should be in parentheses.
Line 40: Grammar/typo
Line 45: What phenomenon? Unclear sentence
Line 47: Not sure what “On the other hand” refers to.
Line 161: precedent: preceding

Reviewer 2 ·

Basic reporting

I find the manuscript to be interesting, but the motivation murky and findings are a bit unclear.

The manuscript can use a bit more editing. I found a misspelling of gender "genender" that should have been picked up by spell checker.

Experimental design

The research question itself is somewhat interesting, but on the whole I don't understand why there would be an expectation that facial deformations would be preferred across people or not. I don't understand why there is a clear gender difference, or what the difference is. The theoretical motivation is lacking.

Validity of the findings

In the supplemental material the authors state that the model is not overfitted, so perhaps the higher order interaction is replicable, but is it understandable?

·

Basic reporting

The manuscript „Unsupervised inference approach to facial attractiveness“ deals with probalistic models that predict the gender of persons that created faces to their maximum attractiveness in virtual face spaces.

The language of the whole manuscript is extremely vague. Therefore, it was quite often hard for me to follow the authors‘ thoughts. E.g., „it has been conjectured that this approach does not capture the complexity of the phenomenon.“ (Abstract). This might be true, but I cannot verify this claim since the authors did not give precise information on why the complexity oft he phenomenon is not captured. Another example: Line 43ff: If you criticize the natural selection hypothesis, please explain your criticism to the reader: „that it does not take into account the phenomenon in its various complex facets“ is no proper critique. Similar passages can be found throughout the manuscript and make it very hard to follow (or to criticize).

To the same point: In my opinion, there is not a sufficient literature review. Quite a few important studies are cited, but the manuscript lacks clarity on the relevant (detailed) results of those studies.

The manuscript needs a language check, preferably by a native speaker. There are numerous misspellings and grammatical errors. Furthermore, I found the citation style rather confusing (only the year oft he citation is in brackets) – this made the manuscript rather difficult to read.

The introduction already includes the main results.

The research question is relevant, especially taking into account that the higher accuracy of the non-linear model includes implications on the nature of visual processing of facial attractiveness. While these implications remain to be tested, the results are in itself quite promising.

Experimental design

From my point of view, the study is well designed. However, methods as well as results are not presented in a way that makes understanding easy for the reader. Descriptions are quite technical without any further explanation.

(1) The Restricted Boltzmann Machine model is not sufficiently explained.
(2) The manuscript lacks a technical comparison between all three methods.

Validity of the findings

I am not sure whether I missed it, but I did not find any information on the number of subjects that were included in the analysis.

The authors state that it should be a next step to test the results with other datasets. I completely agree. However, I think the authors provide enough data for a coherent study.

The high accuracy is surprising and extraordinary.

Additional comments

The strength of the manuscript is that it compares several methods to predict the gender of a person based on their face attactiveness preferences. I like the use of the Maximal Entropy Model in this area of research, especially since it also provides some quantitive insight.

The weakness is the structure of the manuscript itself as well as the language. Especially in the introduction, it is often not clear which point the authors‘ are making exactly. Within the materials, the authors provide lots of detailed information, but I lost track some times since I am not an expert in MaxEntropy.

I’d be highly interested in the effect of sexual orientation on the results, but I guess this information is not available in Ibanez-Berganza et al. (2019).

I’d like to encourage the authors to work on the structure and the language of the manuscript in ordert o make it more accessible. The approach itself is very promising and the results are more than interesting.


Minor points:

Please explain Figure 2 in more detail (longer caption). I did not fully understand what is demonstrated in this figure.

The terms beauty and attractiveness are not distinguished.

Abstract: What are „detailed and global facial features“?

---

## Round 0.2 · accepted · Accept

I'd like to thank you for exhaustively addressing the reviewers' comments, for making the manuscript more accessible to a non-technical audience, and for performing a more thorough editing of language issues.

Reviewer 1 ·

Basic reporting

N/A

Experimental design

N/A

Validity of the findings

N/A

Additional comments

N/A